# Sexual harassment in secondary school: Prevalence and ambiguities. A mixed methods study in Scottish schools

**Helen Sweeting** [ORCID]*, **Carolyn Blake, Julie Riddell, Simon Barrett** [ORCID]¤, **Kirstin R. Mitchell**

MRC/CSO Social & Public Health Sciences Unit, Institute of Wellbeing, University of Glasgow, Glasgow, United Kingdom

¤ Current address: Population Health Sciences Institute, Newcastle University, Newcastle Upon Tyne, United Kingdom
* helen.sweeting@glasgow.ac.uk

**Data Availability Statement:** Data cannot be shared publicly because they are sensitive and based on a small number of schools which, because of their involvement in the Equally Safe at

## Abstract

### Background

Adolescence is characterized by identity formation, exploration and initiation of intimate relationships. Much of this occurs at school, making schools key sites of sexual harassment. Schools often lack awareness and understanding of the issue, and UK research on the topic is scarce. We explored prevalence and perceptions of sexual harassment in a school-based mixed-methods study of 13–17 year-old Scottish adolescents.

### Methods

A student survey (N = 638) assessed past 3-months school-based victimization and perpetration prevalence via 17 behavioral items based on the most commonly used school-based sexual harassment measure ('Hostile Hallways'). Eighteen focus groups (N = 119 students) explored which of 10 behaviors were perceived as harassing/unacceptable and why.

### Results

Two-thirds reported any *victimization*: 64.7% 'visual/verbal' (e.g. sexual jokes) and 34.3% 'contact/personally-invasive' behaviors (e.g. sexual touching; most of whom also reported experiencing visual/verbal types) in the past 3-months. Data suggested a gateway effect, such that contact/personally-invasive behaviors are more likely to be reported by those also reporting more common visual/verbal behaviors. Some survey participants reported being unsure about whether they had experienced certain behaviors; and in focus groups, participants expressed uncertainty regarding the acceptability of most behaviors. Ambiguities centered on behavioral context and enactment including: degree of pressure, persistence and physicality; degree of familiarity between the instigator-recipient; and perception of the instigator's intent. In attempting to resolve ambiguities, students applied normative schemas underpinned by rights (to dignity, respect and equality) and 'knowingness', usually engendered by friendship.

School pilot are potentially identifiable. Data are available from the University of Glasgow MRC/CSO Social & Public Health Sciences Unit Data Access Committee (contact via sphsu-datasharing@glasgow.ac.uk) for researchers who meet the criteria for access to confidential data. Requests will be considered on the basis of their scientific merit and any data release will be subject to a data sharing agreement.

**Funding:** This work was supported by the UK Medical Research Council (https://mrc.ukri.org/) and Scottish Chief Scientist Office (https://www.cso.scot.nhs.uk/) under Grant numbers: MC_UU_00022/3 and SPHSU18 (CB, JR, KM); MC_UU_12017/12 and SPHSU12 (HS); and MC_UU_12017/11 and SPHSU11 (CB, JR, SB, KM). The funders had no role in study design, data collection and analysis, decision to publish, or preparation of the manuscript.

**Competing interests:** The authors have declared that no competing interests exist.

## Conclusions

Our study confirms school-based sexual harassment is common but also finds significant nuance in the ways in which students distinguish between acceptable and harassing. School-based strategies to tackle sexual harassment must engage with this complexity.

## Introduction

In the summer of 2020, a young woman in the UK launched 'Everyone's Invited', a movement which invites survivors to share their stories of sexual harassment and sexual coercion whilst at school or university (https://www.everyonesinvited.uk/). By April 2021, thousands of testimonies had been submitted, with media publicity at that point almost immediately resulting in thousands more. This prompted the UK government to announce an immediate rapid 'Review of sexual abuse in schools and colleges', with a remit including consideration of "the range, nature, location and severity of allegations and incidents, together with context" as well as "what prevents children from reporting" [1]. The resulting review, which reported in June 2021 "revealed how prevalent sexual harassment and online sexual abuse are for children and young people . . . for some children, incidents are so commonplace that they see no point in reporting them" [2].

The Everyone's Invited movement was a shock to many in the secondary education sector. It drew attention to the pervasiveness of the problem, but also the scarcity of UK data surrounding it, and the 2021 UK government review highlighted the need for further research into prevalence, experiences and outcomes [2]. The lack of UK data on school-based sexual harassment and violence contrasts with other high-income countries such as US which have a stronger tradition of research in this area [3]. Our study explored prevalence and perceptions of sexual harassment in a school-based mixed-methods study of 13–17 year-old Scottish adolescents.

Studies of adolescent sexual harassment within school settings are essential for several reasons. Firstly, adolescence is a time of identity formation, exploration with peers and initiation of intimate relationships [4]. Secondly, schools are key sites in which norms are established, including those supportive of a 'rape culture' which normalizes and excuses sexual violence [5, 6], and sexual harassment is enacted [7]. Finally, reviews demonstrate causal associations between any type of victimization at school and both adolescent well-being and later-life health, economic and social outcomes [8].

Since the early nineties, sexual harassment research has focused on measuring prevalence and understanding perceptions of 'what counts' [9]. That this focus has not altered in the intervening 30 years, is at least partly because changes in gender dynamics (progress in equality for women) and attitudes (e.g. 'Me Too' campaign) and new forms of sexual harassment resulting from technological innovations (e.g. online) mean both methodological and substantive aspects of these issues remain important. We review the literature on sexual harassment prevalence and perceptions of 'what counts', before outlining the research questions addressed by our study.

### Sexual harassment prevalence

**Methodological issues.** Measuring sexual harassment prevalence is not straightforward. As many authors point out, "There is no single universal definition of sexual harassment; however, common to all definitions is the fact that the prohibited behavior is unwanted and harms the victim" [10] (p268). Prevalence studies (including of online sexual harassment) report

wide variations in rates. While some variation reflects real differences related to context, much is for methodological reasons. Studies use varying definitions, both in terms of the authors' own perspective and how it is operationalized, varying time-frames and samples with different characteristics [9, 11–16].

Most adolescent studies come from the U.S and have used a modified version of 'Hostile Hallways'. The original questionnaire, developed in the early 1990s, comprises a list of 14 physical/non-physical behaviors, prefaced by a definition of sexual harassment, to which students respond in respect of frequency of victimization and perpetration over their entire school life [17]. Factor analysis indicates that it taps dimensions broadly representing visual/verbal harassment versus sexual contact [18, 19]. Measures of online sexual harassment have also ranged from one or two general items [20, 21] to longer questionnaires, often based on Hostile Hallways [15, 16].

**Substantive findings: Sexual harassment prevalence among adolescents.** As noted above, studies report wide variations in self-reported sexual harassment rates, but all find it is common in adolescence. In both the original 1993 'Hostile Hallways' study of 8-11th grade students and a 2001 repeat survey, around 80% reported they had been the target of, and around 55% had perpetrated, some form of sexual harassment during their school lives; the most common experience was being the target of sexual comments, jokes, gestures or looks [22]. More recent studies based on sub-sets of the original Hostile Hallways items found some form of school-based sexual harassment was reported by 48% of 7-12th graders in respect of the 2010–11 school year [23], and by 43% of 9th graders in respect of the past three months [24]. Internet harassment victimization was reported by 35% and perpetration by 21% of 10–15 year-olds participating in a 2006 US national cross-sectional online survey [25].

Although there are some exceptions (e.g. [26]), these and similar studies of adolescents generally find higher rates of victimization among girls and of perpetration among boys. Victimization and, some studies also find, perpetration, are also higher among young people whose sexual orientation is not heterosexual and who are not cis-gender [15, 27–29]. Rates also vary by age/grade, most studies finding highest levels around 9th grade (age 14–15) [17, 26, 30]. There is also evidence of an association between sexual harassment victimization and perpetration in adolescence [23, 31].

There have been almost no published UK-based prevalence studies of adolescent sexual harassment, particularly within the academic literature or focusing on school-based experiences (although there have been recent studies of rates of adolescent partner interpersonal violence (e.g. [32]) and related school-based interventions (e.g. [33]). A 2010 online poll of 788 UK 16–18 year-olds about school-based sexual harassment found 71% had heard sexual name-calling towards girls a few times a week or more, 28% had seen sexual pictures on mobile phones a few times a month or more and 29% of girls had experienced unwanted sexual touching [34]. A 2014 largely online survey of 13–21 year-old young women found 59% had experienced past year school/college-based sexual harassment, among whom 37% reported jokes or taunts of a sexual nature and 20% had experienced unwanted sexual attention [35]. The sampling, opt-in and online nature of both studies means their results may not be generalizable. A 2017 school-based survey which asked how often sexual harassment was experienced in school and elsewhere found 9.5% of English 12–14 year-olds reported any experience, however results were not disaggregated by setting [33]. The 2021 UK government review found that among 13–18 year olds, 92% girls and 74% boys reported sexist name-calling happened 'a lot' or 'sometimes' between people of their own age, while among girls, 88% had been sent pictures/videos they did not want to see and 64% had experienced unwanted touching, with increased likelihood of both offline and online sexual harassment and violence among older adolescents [2].

### Research on what behaviors are perceived as sexual harassment

**Methodological issues.** The term 'sexual harassment' has been defined by many jurisdictions and organizations, including the US Centers for Disease Control (for gathering public health surveillance data) [36] and the UK Equality and Human Rights Commission (as employer guidance) [37]. However, these definitions are unlikely to be known to people without specialized or professional knowledge, particularly children and young people. Sexual harassment research has adopted both a priori definitions (based on general statements about the type of behavior or a list of specific behaviors) and empirical definitions (based on asking respondents whether they have ever been sexually harassed and analyzing/classifying the experiences described) [38]. Some authors have criticized the idea that it should include any behavior which makes someone feel uncomfortable [39] and subjective perceptions may present dilemmas when ambiguous behaviors are inconsistently perceived as sexual harassment [10]. This may be a particular issue for adolescents; pressure to develop sexual relations at a time when they are learning to express their desires appropriately may mean it is hard to differentiate playful from abusive behaviors [18, 40, 41]. Thus, qualitative research with UK adolescents has found confusion as to the boundaries between consent and coercion [42], and what constitutes sexual violence or exploitation [43].

**Substantive findings: What 'counts' as sexual harassment among adolescents?.** Studies almost universally find differences in what 'counts' as sexual harassment according to characteristics of both the behavior and context, and of the individual making the rating. In respect of the behavior, a 1983 study of US undergraduates found incidents initiated by young women were viewed more positively [44]. A 1995 study of US high school students found criteria on what was viewed as sexual harassment included use of quid pro quo exchanges, physical force, repetition, levels of coercion and perpetrator intent, whether the target was uncomfortable, status/power differences and familiarity between the two people [45]. It is possible that some of the factors identified in these older studies may have changed over time. While one study comparing undergraduates in 1990 and 2000 found "surprisingly few cohort differences in the perception of harassment" [46] p344, another, comparing data gathered before and during the #MeToo movement found past unwanted experiences were increasingly recognized as 'sexual assault' over time [47].

In respect of online compared with offline, comments of a misogynist nature, relating to dress or using nicknames were seen as having more impact online than in classroom settings among US undergraduates [48]. In a Belgian study, online sexual harassment was perceived as more severe by adolescents when personally targeted, hard to escape (e.g. anonymous perpetrator, public message), involved insults, included non-consensual sex-related image-sharing or was frequent and adult-initiated [49], while a UK study of sexting (sharing sexual messages/images) found little concern about sexual approaches from strangers, who could be dismissed as 'weirdos', but far more about sexting from peers [50].

Rater gender is also important, with women tending to view a greater range of behaviors as sexual harassment [51]. However, while some studies have found young women are more likely to define more ambiguous situations as sexual harassment and/or particular behaviors as more severe or unacceptable [2, 45, 52], others have not [53, 54].

## This study

Our mixed-methods study thus sought to meet a gap in prevalence data for the UK, and to qualitatively elucidate contemporary perceptions and understandings of sexual harassment from the perspective of in-school adolescents. In doing so, it adds to understanding of the ambiguities in what counts; this ambiguity is one of the key reasons that school-related

incidents may go unreported [2]. We included both online and offline sexual harassment, given the convergence of adolescent online and offline interactions [55] and the fact that, within adolescent social groups, harassing behaviors often cut across the two contexts [56]. Reflecting its mixed methods, our reporting follows both Strengthening the Reporting of Observational Studies in Epidemiology (STROBE) and Standards for Reporting Qualitative Research (SRQR) guidelines [57, 58] (S1 and S2 Checklists).

We collected survey data to address three prevalence-related questions:

- What is the self-reported prevalence of sexual harassment (victimization and perpetration) among adolescents?

- What is the relationship between sexual harassment victimization and perpetration?

- How do sexual harassment victimization and perpetration differ according to gender, sexual orientation and age?

We also collected qualitative (focus group) data to address three questions related to perceptions:

- Are behaviors used to measure prevalence of sexual harassment in surveys also understood/ labelled by young people as 'sexual harassment'?

- What 'counts' as sexual harassment; what do young people consider acceptable and unacceptable behavior?

- What influences whether young people perceive a behavior to be acceptable or not?

## Methods

### Sample

Data were drawn from students in three Scottish secondary schools that agreed to participate in a pilot of a whole school approach to end gender-based violence (Equally Safe at School) implemented by the non-governmental organization, Rape Crisis Scotland (see https://www. rapecrisisscotland.org.uk/equally-safe-at-school/ for further information). Students completed an online survey and some subsequently participated in focus group discussions prior to intervention implementation. The schools varied in respect of size, location (two drawing most students from large urban areas, one mostly from a small town) student socio-economic status (indicated by proportions receiving free school meals, with one above, one roughly equivalent to and one below the Scottish school average) and academic achievement [59].

**Online survey sample.**   In two of the three schools, all students in Scottish school years Secondary 2 (S2—average age 13), S4 and S6 (average ages 15 and 17) were invited to participate. In the third school, only those in years S4 and S6 were invited, because the school considered the topic unsuitable for the S2 age-group. We provided information sheets (including opt-out consents) for parents, separate student information sheets, and both verbal explanation and a tick-box consent at the start of the survey (information and consent forms available in S1 File).

The potential sample was 904 students, of whom 638 (70.6%) completed a survey between June and November 2019. Opt-out was very low, with almost all non-response due to student absence on the survey day or, in one school, missing the survey due to organizational issues (planned additional data collection not completed due to COVID-19 related school closure). The achieved sample comprised N = 281 (44.0%) boys and N = 340 (53.3%) girls (N = 17 (2.7%) missing on this variable); there were N = 178 (27.9%) students from the S2 school year,

N = 305 (47.8%) from S4 and N = 155 (24.3%) from S6. (Note that S6 year groups are smaller since around 40% of Scottish students leave secondary school after S4 or S5, the majority entering Further Education (that is post-secondary school study not taken as part of an undergraduate or graduate degree).)

**Focus group sample.** Focus group discussions were conducted in January-February 2020 with a single class group of students from each of the three school years in two schools (not the third due to COVID-19 related school closure). Further information sheets were provided to parents and students. These described us as university researchers studying the Equally Safe at School project and inviting the student to join a small group discussion, some of which "will cover sexual harassment at school which you may find uncomfortable. We will be discussing issues in general, and not personal experiences. We need to ask these questions to understand the scale of the problem . . .". The invitation explained that participation was voluntary and described our procedures in respect of confidentiality, anonymity and consent. Participants provided written consent at the start of each group (information and consent forms available in S1 File). Each participating class was divided into three (single gender if possible) groups, numbers ranging 4–10, with most students choosing their group. The total number of students participating in a focus group was 119 (41 S2s, 46 S4s and 32 S6s), split into 18 groups (three girl-only, two boy-only and one mixed S2 groups; three girl-only, two boy-only and one mixed S4 groups; three girl-only and three boy-only S6 groups).

## Procedures

**Online survey.** The online survey was completed in school under exam conditions led by researchers and survey assistants, via students' phones or tablet computers provided for those without a (suitable/working) phone. In addition to sexual harassment, it included items on socio-demographic factors, including gender and sexual orientation (heterosexual or straight; gay or lesbian; bisexual; other).

*Sexual harassment victimization and perpetration* were assessed via 17 behavioral items (Table 1): the original 14-item Hostile Hallways questionnaire [17] with slight modifications to encompass the possibility that the behavior might be online (e.g. "Shown, given, or left sexual pictures. photographs. illustrations, messages or notes" rephrased as "Showed you or sent you sexual images or messages that you did not want to see") and three additional items (Table 1, items 15–17) representing personally-targeted online behaviors included in other adolescent studies [16, 25] (e.g. "Pressured you to send them a naked (nude) or sexual picture of yourself"). The test-reliability correlation coefficient for the original Hostile Hallways victimization behaviours has been assessed as 0.95 [22].

While mindful of suggestions that the words 'sexual harassment' might act as cues when assessing prevalence [9], we followed the Hostile Hallways methodology which includes a definition [17, 22, 23]. The items were therefore preceded by "In this section there will be questions about your experience of sexual harassment within the school, but also on the way to school. What is sexual harassment? It is any unwanted behavior of a sexual nature that you find offensive or which makes you feel uncomfortable, intimidated or humiliated".

To assess *victimization*, students were asked "In the last 3 months, how often, if at all, has anyone done any of the following things to you at school or on the way to or from school when you did not want them to? (This includes students, teachers, other school employees, or anyone else.)" To assess *perpetration*, students were asked "In the last 3 months, how often, if at all, have you done any of the following things to someone else at school or going to school when they did not want you to?" in respect of the same items. Response options for both

**Table 1. Individual victimization and perpetration items by gender.**

| Survey items | Ever victim (past 3 months) | | | Ever perpetrator (past 3 months) | | |
|---|---|---|---|---|---|---|
| | Boys % | Girls % | Chi-sq sig | Boys % | Girls % | Chi-sq sig |
| 1-Made sexual jokes, gestures or looks (V/V)# | 52.7 | 48.7 | 0.356 | 37.5 | 14.8 | <0.001 |
| 2-Showed you or sent you sexual images or messages that you did not want to see (V/V) | 17.1 | 32.9 | <0.001 | 4.0 | 1.8 | 0.106 |
| 3-Wrote sexual messages / graffiti about you in public places (eg. on toilet walls, in changing rooms) (V/V) | 8.5 | 7.9 | 0.789 | 4.0 | 0.9 | 0.012 |
| 4-Spread sexual rumours about you online or in person (V/V) | 15.1 | 20.0 | 0.134 | 6.4 | 2.7 | 0.030 |
| 5-Said you were gay or lesbian, in a hurtful way (V/V) | 26.3 | 19.6 | 0.056 | 15.9 | 2.7 | <0.001 |
| 6-Spied on you as you dressed or showered at school (PC/P)# | 3.8 | 3.5 | 0.868 | 0.00 | 0.00 | n/a |
| 7-Flashed /'mooned' at you (showed their private parts or exposed themselves) (V/V) | 18.6 | 10.8 | 0.008 | 8.4 | 1.5 | <0.001 |
| 8-Touched, grabbed, or pinched you in a sexual way (PC/P) | 19.4 | 16.6 | 0.386 | 7.7 | 1.2 | <0.001 |
| 9-Brushed up against you in a sexual way on purpose (PC/P) | 15.2 | 11.6 | 0.207 | 3.2 | 1.2 | 0.087 |
| 10-Pulled at your clothing in a sexual way (PC/P) | 8.3 | 7.3 | 0.668 | 3.2 | 0.6 | 0.023 ~ |
| 11-Pulled off or down your clothing (PC/P) | 5.5 | 4.2 | 0.490 | 3.6 | 0.3 | 0.003 ~ |
| 12-Blocked your way or cornered you in a way that made you feel uncomfortable (PC/P) | 10.7 | 14.7 | 0.154 | 2.0 | 0.0 | 0.013 ~ |
| 13-Made you kiss him / her (PC/P) | 6.3 | 5.8 | 0.794 | 1.6 | 0.0 | 0.032 ~ |
| 14-Made you do something sexual other than kissing (like touching their private parts) (PC/P) | 5.5 | 2.4 | 0.052 | 0.4 | 0.0 | 0.429 ~ |
| 15-Taken a picture to see under your clothes, eg. up your skirt or down your shirt (PC/P) | 2.8 | 1.2 | 0.226 ~ | 0.8 | 0.0 | 0.184 ~ |
| 16-Forwarded a naked or sexual picture of you to others, without your agreement (PC/P) | 3.5 | 5.5 | 0.265 | 0.8 | 0.6 | 1.000 ~ |
| 17-Pressured you to send them a naked (nude) or sexual picture of yourself (PC/P) | 5.0 | 14.3 | <0.001 | 1.2 | 0.0 | 0.078 ~ |

# V/V = visual or verbal; PC/P = physical contact or personally-invasive

~ Fisher's exact test used when small expected cell counts.

victimization and perpetration were 'often', 'occasionally', 'rarely', 'never' (as in the original Hostile Hallways) and 'not sure' (added to reflect our interest in understandings).

**Focus groups.** Each focus group discussion was facilitated by one researcher; all authors facilitated one or more groups; all had previous experience of qualitative data collection from young people and/or in respect of sensitive topics. Teachers were present but not involved. The three groups per class were held simultaneously in large rooms (e.g. school library, large classrooms) and audio-recorded with consent. They began with a whole-class introduction describing our research and explaining the consent forms and discussion ground rules. Students were asked to speak about their perceptions ("what you think"; "your ideas") but avoid sharing personal experiences, naming others, detailing events or repeating any of the discussion to others outside the group. They were also told they could opt-out of the discussions at any time, without any questions asked.

Almost all the qualitative data presented here were obtained via a discussion focused around what counts as sexual harassment; for which each group was provided with a large sheet of paper with three columns, 'OK', 'not OK' and 'not sure' and packs of ten cards each listing a behavior (Table 2). Seven cards reflected items from our survey. Four were modified (by removing: "in a sexual way" from both "Touched, grabbed or pinched you" and "Pulled at your clothing"; "that you did not want to see" from "Showed you or sent you sexual images or messages"; and "in a hurtful way" from "Said you were gay or lesbian") to draw out discussion of what students thought made behaviors both sexual or not sexual, and OK or not OK. Three items were added (e.g. "Giving gifts of a sexual or romantic nature") to focus on less clear-cut examples and allow for differences of opinion to emerge. Students were not provided with a definition of sexual harassment, or told the cards referred to this. Students took turns to pick a

**Table 2. Discussion prompts (with number of equivalent survey item).**

| Survey item number | Prompt |
|---|---|
| 1. | Made sexual jokes |
| 2. | Showed you or sent you sexual images or messages |
| 5. | Said you were gay or lesbian |
| 8. | Touched, grabbed or pinched you |
| 10. | Pulled at your clothing |
| 13. | Made you kiss him / her |
| 17. | Wanted you to send them a naked nude) or sexual picture of yourself |
| New | Staring in a sexually suggestive manner, or whistling |
| New | Making sexual comments about appearance, clothing or body parts |
| New | Giving gifts of a sexual or romantic nature |

card at random and were prompted (if necessary) to discuss whether the behavior was OK and what might make it more/less OK. Additional data in respect of how young people understood/labelled these behaviors were obtained at the start of a second activity, based on 'soft systems' methods [60] and devised to help students think about wider systemic influences on sexual harassment. Key findings from the activity will be reported separately, but here we used data from the first step, in which students were probed for their own overall definition/terminology for the behaviors described on the cards.

## Permissions

The study received permission from University of Glasgow College of Social Sciences ethics committee (#400180040; #400180236), schools' local authorities and head teachers.

## Analysis

**Online survey.**　Quantitative analysis of the survey items was conducted via SPSS.25 and based on those providing valid responses to relevant items. Individual behavioral items are presented via descriptive statistics for each item (victimization and perpetration), and analyses of composite measures representing experience/perpetration of items representing: 'visual/verbal' and 'physical contact/ personally directed or invasive' (hereafter 'contact/personally-invasive') victimization or perpetration, based on previous factor analyses of Hostile Hallways items [18, 19, 61] and as indicated on Table 1. The six 'visual/verbal' items included behaviors such as "Made sexual jokes, gestures or looks", "Showed you or sent you sexual images or messages that you did not want to see" and "Spread sexual rumors about you online or in person"; the 11 'contact/personally-invasive' items included behaviors such as "Touched, grabbed, or pinched you in a sexual way", "Made you kiss him/her" and "Pressured you to send them a naked (nude) or sexual picture of yourself". We used crosstabulations (chi-square tests) to examine relationships between sexual harassment victimization and perpetration and differences according to gender, sexual orientation and age. Note that our quantitative analysis in respect of gender was based on the question what "was written on your birth certificate". While recognizing rates of sexual harassment are higher among young people who are not cisgender [15], we did not include current gender identity because the numbers reporting this were very small in the survey (transgender N = 4, non-binary or other N = 10). In addition, this information was not collected for the focus group discussions. Thus in this paper, we refer to two gender categories, girl/woman and boy/man.

**Focus groups.** Audio recordings were transcribed; a few combined facilitator-expanded notes with transcription (where audio quality was poor). Analysis followed a thematic approach [62]. Following data familiarisation and initial discussions (all authors), we used an inductive approach to generate initial core codes as follows: Friendships/relationships; Familiarity; Intentions and 'received' meanings; Social context; Physicality; Pressure and persistency; Gender; Recognition of rights and values). These (and sub-codes) were coded via NVivo by three authors (CB, JR, SB). The resulting NVivo qualitative data analysis software output was again discussed (by CB, KM, HS) in the process of generating the themes presented here and during write- up.

## Results

### Quantitative (online survey) results

**What is the self-reported prevalence of sexual harassment (victimization and perpetration) among young people?.** Table 1 shows frequencies of each behavioral item (ever victim and perpetrator versus 'never', with 'not sure' coded as missing) according to gender. Overall, by far the most frequently experienced behavioral item (i.e. as a *victim*) in the past three months, reported by around half the sample, was being the recipient of sexual jokes, gestures or looks; around a quarter had been shown unwanted sexual images or messages and a fifth had been described as gay or lesbian in a hurtful way. In contrast, several items were only very rarely reported, including having had a picture taken to see under clothes; being made to do something sexual other than kissing; a naked/sexual picture of themselves being forwarded without consent; and having clothing pulled off or down. S1 Table shows the proportions responding often/occasionally, rarely, never and not sure to each item. The proportions responding 'not sure' were striking, and varied between items, suggesting (sometimes clearly justified) uncertainty, rather than a group consistently choosing this answer as a way of not responding. For instance, while around 9% were 'not sure' whether they had been the targets of sexual gestures, jokes or looks and 8% whether others had spied on them while showering or getting dressed, only 2–3% were 'not sure' whether they had been pressured to send a naked picture or whether their clothing had been pulled off/down.

The frequency order in which the behavioral items were experienced as a victim was very similar to that of reported perpetration, although rates of perpetration were markedly lower (Table 1 and S1 Table). Thus, while making sexual jokes, gestures or looks was reported by 24% and calling someone gay/lesbian by 8%, fewer than 1% reported: blocking or cornering; making someone kiss them; forwarding a naked/sexual picture; pressuring someone to send a naked/sexual picture; spying; making someone do something sexual other than kissing; or taking a picture up someone's clothes. In contrast to the victimization items, 'not sure' was consistently reported by around 2–3% in respect of each perpetration item.

**Relationships between sexual harassment victimization and perpetration, and differences according to gender, sexual orientation and age.** Table 1 shows significant gender differences for only three victimization items. "Showed you or sent you sexual images or messages that you did not want to see" and "Pressured you to send them a naked (nude) or sexual picture of yourself" were more likely for girls and "Flashed /mooned at you (showed their private parts or exposed themselves)" for boys. Gender differences for two further items approached significance, both more likely for boys: "Said you were gay or lesbian, in a hurtful way" and "Made you do something sexual other than kissing (like touching their private parts)". Rates of all perpetration items were markedly higher among boys, most of these differences were statistically significant. The behaviors with the smallest gender differences were "Made sexual jokes, gestures or looks" (37.5% boys, 14.8% girls) and "Spreading sexual rumours" (6.4% boys and 2.7% girls).

S2 Table shows composite prevalence for any visual/verbal behaviors and any contact/personally-invasive behaviors. In respect of *victimization*, it shows that 64.7% reported any of a visual/verbal type, 34.3% any of a contact/personally-invasive type and 68.3% either type. In respect of *perpetration*, 29.0% reported any visual/verbal, 6.4 any contact/personally-invasive and 30.0% either type. Further analyses (S3 Table) showed that for both victimization and perpetration, contact/personally-invasive behaviors (i.e. those which would generally be regarded as more severe) were only very rarely reported in the absence of visual/verbal behaviors. Thus, among those with valid data in respect of both types of *victimization* (N = 525): 33.9% reported neither type; 29.7% visual/verbal only; 2.7% contact/personally-invasive only; and 33.7% both. In the same way, among those with valid data (N = 551): 71.7% reported neither *perpetration* type; 21.81% visual/verbal only; 0.2% contact/personally-invasive only; and 6.3% both. There was also a very strong association between victimization and perpetration (S4 Table): almost all (94.7%) of those who reported no victimization also reported no perpetration and none reported contact/personally-invasive perpetration. However, contact/personally-invasive perpetration was reported by a fifth (19.5%) of those also reporting contact/personally-invasive victimization.

Fig 1 shows composite victimization and perpetration variables (categorised none; visual/verbal only; any contact/personally-invasive) according to gender, sexual orientation and school year group. Reflecting the Table 1 results, it demonstrates that while there were no gender differences in either victimization type, rates of perpetration, particularly contact/personally-invasive types were higher among boys. Experience of contact/personally-invasive (but not visual/verbal only) harassment was higher among students whose sexual orientation was not heterosexual. Finally, rates of both victimization and perpetration increased with age, with jumps in the rates of contact/personally-invasive types between S2 (age 13) and S4 (age 15).

## Focus group results

We begin by briefly discussing the language that students used to label sexual harassment behaviors, and then explore what students thought 'counted' as sexual harassment, and as acceptable versus unacceptable behaviors, and in what context. All quotes are attributed to a group coded according to school, year group and gender (e.g. Sc1_S2G.2 = School 1, S2 year group, the second girl-only group): individual students are not differentiated in any exchanges because of difficulties consistently distinguishing them during transcription.

**Are behaviors used to measure prevalence of sexual harassment in surveys also understood/labelled by young people as 'sexual harassment'?.** Following the card-sorting task—and having *not* been explicitly provided with the term 'sexual harassment' for the behaviors they had just discussed—students formulated a wide range of terms to label them. Three of the 18 groups decided they were 'sexual harassment'. Some of the others included 'harassment' (only) or incorporated 'sexual'(e.g. "sexual behaviors"; "sexual abuse"; "sexual assault"; "sexual issues or actions"). However, around half the groups used neither 'sexual' nor 'harassment', instead, choosing terms such as "inappropriate behavior", "things that have consequences", "abuse", "unwanted", "non-consensual" or behaviors meaning "people might feel pressure or uncomfortable". Ideas of non-consent and discomfort were more in evidence among S4 and S6 (i.e. older) students.

**What do young people consider acceptable and unacceptable behavior?.** The 'OK', 'not sure' and 'not OK' cards task suggested the ten behaviors could be ordered on the basis of increasing acceptability. This did not appear to be clearly patterned on the basis of gender, year group or school.

## VICTIMIZATION

## PERPETRATION

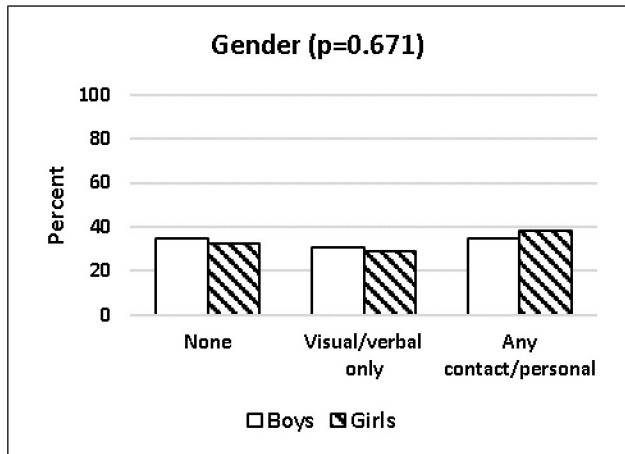

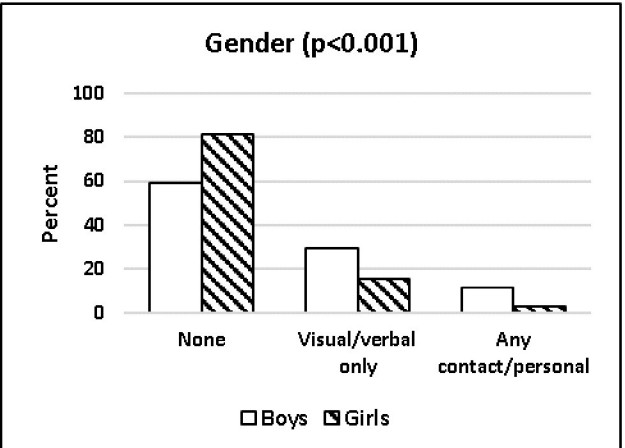

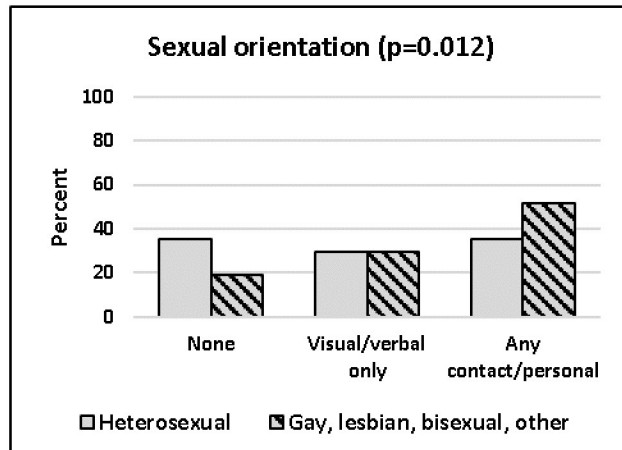

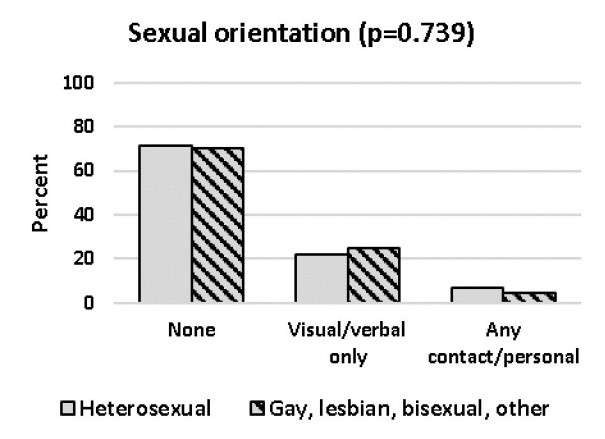

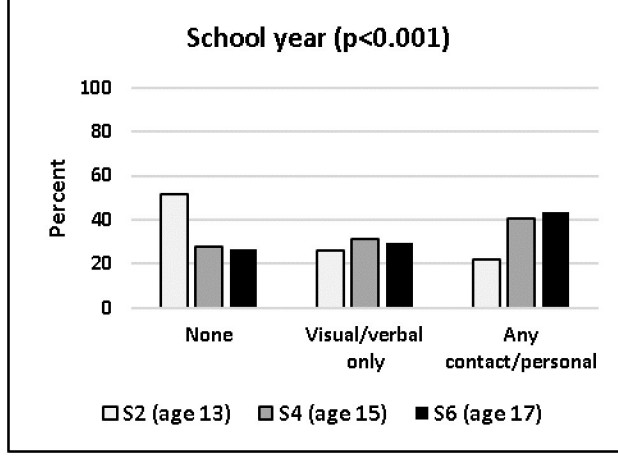

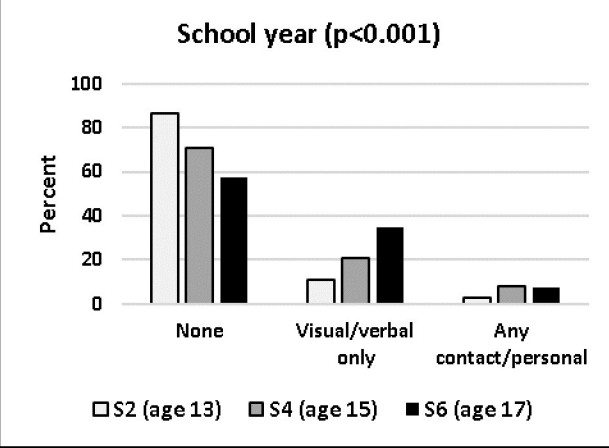

**Fig 1. Composite victimization and perpetration variables (none, visual/verbal only; any contact/personally-invasive) according to gender, sexual orientation and school year group (with significance of group differences).**

Table 3 shows the least acceptable behaviors were "Made you kiss him/her" and "Wanted you to send them a naked (nude) or sexual picture of yourself". Both 'made' and 'wanted' were associated with being forced to do something which was 'not OK'. These were followed by "Staring in a sexually suggestive manner, or whistling" and "Showed you or sent you sexual images or messages". No group suggested any of these behaviors was 'OK'.

Increasingly more acceptable and more ambiguous were "Making sexual comments about appearance, clothing or body parts"; "Said you were gay or lesbian", "Pulled at your clothing", "Touched, grabbed or pinched you" and "Made sexual jokes"; half or more of the focus groups were unsure about the acceptability of each of these. Finally, only one group regarded "Giving gifts of a sexual or romantic nature" as 'not OK' and most were unsure. Students generally felt this would be unusual in a Scottish context and more something that happened in American movies. All six of these behaviors prompted considerable discussion about the importance of context and nuances around the exact nature of the behavior.

**What influences whether young people perceive a behavior to be acceptable or not?.** Discussions around the acceptability of each behavior highlighted several influencing factors. Central were the characteristics of the behavior, including degrees of pressure, persistency and physicality, based on whether individual rights were infringed. Distinct from the behavior itself were the degree of familiarity and relationship between instigator and recipient and the instigator's (perceived) intent. Gender was also important; both whether behaviors were within or between genders and differences in what was regarded as acceptable for adolescent boys compared with girls. Each of these factors is explored below.

*Characteristics of the behavior*: *Pressure, persistency and physicality*. Pressure and/or persistency were almost always described as making a behavior unacceptable. Students condemned any form of pressure, whether physical or social/emotional and regardless of whether or not it was in the context of a close relationship. Discussions around "Made you kiss them", the least acceptable of the ten behaviors, particularly highlighted these views:

*No—if you're making them then I don't think it's right.*

*Cos it's against their will.*

(Sc1_S4G.2)

A girl-only group highlighted the potential for aggression:

**Table 3. Numbers of focus groups (total N = 18) deciding each behavior was 'OK', 'not OK' or that they were 'not sure'—behaviors ordered according to increasing perceived acceptability.**

| Behavior | OK | Not sure | Not OK |
|---|---|---|---|
| Made you kiss him / her | 0 | 1 | 17 |
| Wanted you to send them a naked (nude) or sexual picture of yourself | 0 | 4 | 14 |
| Staring in a sexually suggestive manner, or whistling | 0 | 5 | 13 |
| Showed you or sent you sexual images or messages | 0 | 6 | 12 |
| Making sexual comments about appearance, clothing, or body part | 0 | 9 | 9 |
| Said you were gay or lesbian | 1 | 9 | 8 |
| Pulled at your clothing | 1 | 9 | 8 |
| Touched, grabbed, or pinched you | 0 | 12 | 6 |
| Made sexual jokes | 0 | 13 | 5 |
| Giving gifts of a sexual or romantic nature | 3 | 14 | 1 |

*My mum always says 'they can't make you', but see, they can, they really can make you. See if you are sitting there, they can grab you.*

*It can be in an aggressive way.*

(Sc2_S2G)

While a boy-only group described social/emotional pressures:

*I'd say there is an element of mental toying with you if it's making you like that.*

*If you are playing those games like truth or dare—that can be psychological, like peer pressure.*

(Sc1_S6B)

Persistent or repeated requests were also described by all participants as making a behavior unacceptable.

*I think worse like if it's over a long time, like if they continue to persist.*

*[...]*

*If you say no the first time they shouldn't continue asking.*

(Sc1_S4G.2: Wanted you to send a sexual picture)

The acceptability of touched, grabbed or pinched depended on "*what way it is, so if it's just going up to someone and, like, touching them, or something, you do that all the time with your pals, [...] so I think it's alright*" (Sc2_S6B.1). Touching someone else was generally perceived as only acceptable between friends, among same-gender groups, and (unless in a close, consenting relationship) if it was not on an intimate part of the body. Thus, "*if they're pinching their arm and that, that's fine. But, like, touching and grabbing other places isn't really right*" (Sc1_S4G.1). If touching was sexual then it was only OK with consent, otherwise "*that's abuse . . .bang out of order*" (Sc2_S2Mx).

Underpinning discussions around behaviors involving pressure, persistence and physicality were students' understandings about rights, both legal and in terms of individual rights to have one's feelings respected. Consent was crucial to acceptability; a behavior was considered unequivocally 'not OK' if the group perceived it as being forced:

*Made you? Made you! No that is not ok! You can't make someone kiss someone.*

*That's horrible.*

*Let's say you asked someone to kiss you, and they say no, and you kiss them anyways, that's wrong.*

(Sc2_S2G)

There was awareness of laws around sharing sexual images, but not all were clear on their detail, which impacted on understandings of whether this was OK from a legal point of view (note that in the UK it is an offence to make, distribute, possess or show any indecent images of anyone aged under 18, even if the content was created with the consent of that young person). A group of S4 students (aged about 15) all agreed this was not allowed "*aye we're too young*" (Sc1_S4B). However, some S6s (aged about 17) were (incorrectly) very clear that being

"*of age*" meant from age 16 (Sc1_S6G.1) and others were unsure whether it was allowed if aged under 18, whether consent was required and whether it mattered if the images were of themselves or another (Sc2_S6B.2).

However, separate from legal rights, students also more frequently used "un/comfortable" in respect of how a behavior was received, suggesting recipient feelings, bodily autonomy and privacy as more important criteria for acceptability:

> *If someone's uncomfortable with it, and it happens, then it's obviously not alright.*
>
> (Sc2_S6B.1: Touched, grabbed or pinched)

*Familiarity and relationship between instigator and recipient.* No (or limited) familiarity with the instigator was seen as a reason to regard a behavior as inappropriate. To illustrate this, students often used the case of "an old man', "stranger" or "random person", the latter referring to the idea that the behaviors were both unexpected and out of context. (Hypothetical) examples were therefore generally out of school and regarded as "weird" or "creepy".

> *[Researcher] Would it matter who sends it to you?*
>
> *If it's an old guy. Think about that.*
>
> *Yeah.*
>
> *[Researcher] Would it ever be ok?*
>
> *No.*
>
> (Sc1_S2G.1: Showed/sent you sexual messages)

> *If it's [named friend (girl)] it's OK but if it's some weird man coming up to touch you that's not OK.*
>
> (Sc2_S2Mx: Touched, grabbed or pinched)

Some lower-level behaviors were acceptable within friendship groups. Generally, knowing the person well meant there was trust among individuals: *"if it's with a friend, and you are used to it"* (Sc2_S4Mx). However, making sexual jokes and calling someone gay or lesbian if you were unsure how they might react was not OK: "*You don't just go up to someone and start saying stuff. Cause they might take it wrongly*" (Sc2_S4Mx). This meant that, while acknowledging these sorts of jokes could cause offence if in public, two boy-only discussion groups suggested they were more acceptable on PlayStation chat among trusted friends, where *"we know we are having a laugh and it is private*" (Sc2_S2B; Sc2_S4B). Being a friend might also make something feel more acceptable. A group of boys discussing pulling at clothes commented that although *"there could be a sexual manner to it"* (Sc2_S2B), doing this to a friend, especially another boy would mean it was more likely to be seen as a joke, while a girl-only group member, talking about boys' hands on her waist suggested that "*I suppose it does just determine how close you are with the person, if you're comfortable with it or not. . .*" (Sc1_S4G.1). Friends had shared understandings and should therefore 'know' that an otherwise potentially unacceptable behavior was meant as a joke: "*If it's one of your friends and they understand. If it's an actual joke and, just kidding, by the way, that's alright*" (Sc1_S6G.1: Made sexual jokes). Friendship also meant that the recipient should feel comfortable about providing feedback on the basis it would be acted upon: "*If it's your mate they'd understand, they'd say just don't do it again*" (Sc1_S4B: Made sexual jokes).

More sexual behaviors were acceptable within a close relationship, but still required care. Making sexual comments was OK "*if you are in a relationship and you know the person and their boundaries*" (Sc2_S6B.2), as was staring at someone "*if they are in a relationship, fair enough, if that person feels comfortable with it*" (Sc1_S4B). Similarly, in respect of being shown or sent sexual material:

*I feel like there are circumstances where sometimes people might be in a relationship and sometimes that's. . .*

*Aye, then it's fine for it, but if you're not then. . .*

(Sc1_S4G.1)

However, not all were sure about this; a girl from another group suggested that "*If it's a good boyfriend, he wouldn't ask this*" (Sc1_S2G.1).

Flirting, which might, or might not, lead to a close relationship also required care. A group discussing touching, grabbing or pinching were questioned:

*[Researcher] What if they were already flirting?*

*You can cross the line a wee bit.*

*But you could nae [not] go right to the other end, like line, don't cross it.*

(Sc1_S4B)

However, this was a tricky balance since there was little or no explicit discussion between those in a (potential) relationship about boundary-setting. When one group was asked about this, they suggested their decisions as to the acceptability of behaviors within relationships were on the basis of 'knowing' the person and 'feeling' it was OK, "*they are virtually never set by a discussion, they are from experience or a person reacts and says 'that is too much'*" (Sc2_S6B.2: Making sexual comments).

*Instigator's (perceived) intent.* Acceptability of a behavior was also related to both intentions and received meanings. These could be broadly grouped as comedic, positive (e.g. compliments) and harmful (e.g. objectifying).

Behaviors were unacceptable if the (perceived) intention was to cause distress. Thus, while calling someone gay or lesbian as "*a bit of banter [teasing, joking]*" (Sc1_S4B) was acceptable, doing so "*in a hurtful way, that is not ok, especially in our day and age*" (Sc1_S2B), similarly, if "*they've done it to just make them feel uncomfortable, then no [not OK]*" (Sc1_S6G.1: pulling clothing). Related to this, targeted behaviors were therefore unacceptable, "*I think if you mention a name it's a lot worse*" (Sc1_S4G.1: Made sexual jokes). This had particular implications in respect of calling someone gay or lesbian, since that person might be questioning their sexuality or keeping their sexual identity hidden, and might be distressed by having attention drawn to this:

*Or if someone's self-conscious about it, then it could be considered aggressive.*

*Yeah. So they, themselves. . .*

*Are maybe thinking, they are, and that. Or maybe they're like considering if they are or not as well, and they're not sure, and then someone says that to them, then they might be, they might think, oh maybe I am.*

(Sc2_S6B.1)

There was also some discussion around the importance of responsibility. This could be with regard to responsibility for any unintentional harms *"with a joke—you know what I mean —I think as long as you're willing to apologize after that if you've like offended someone I think it's OK"* (Sc1_S6G.2). However, it could also make *"just sharing and forwarding"* sexual jokes more acceptable because *"it is not us making the joke, it is other people"* (Sc2_S2B).

As highlighted above, perceived comedic intentions meant certain behaviors were acceptable within friendship groups. In part, this was because of a mutual assumption that good friends would not want to hurt each other. Among friends, saying someone was gay or lesbian in the context of a funny conversation or when you knew they were heterosexual, should be perceived as a joke. However, mistakes were always possible:

> *I suppose you could be saying it in like a joking way to a friend, but they might be uncomfortable.*
>
> *You'd be better off not saying it.*
>
> *Yeah.*
>
> (Sc1_S6G.2: Said you were gay/lesbian)

As the above quote illustrates, more important than the instigator's *actual* intentions, perceived intentions depended, in an almost circular way, on how the recipient interpreted, and therefore *felt*, about a behavior:

> *If it's just. . .if it's not made in, like, a sexual way, then it's not that bad. But again it just depends if they feel comfortable or not. But if it's done in, like, an angry or aggressive way it's not okay.*
>
> *[Researcher] What would make you say that it's not in a sense in a sexual way or in a way that makes . . .*
>
> *Uncomfortable. . .if it made me uncomfortable.*
>
> (Sc1_S4G.1: Touched, grabbed or pinched)

This could mean differences between individuals in how acceptable they found certain behaviors:

> *Some people might take it as a joke but some others. . .*
>
> *Some are too sensitive.*
>
> (Sc2_S2Mx: Pulled at clothing)

These intention-related factors meant that some young people thought it could be difficult to determine where the line was located, as discussed by a group of boys in respect of making sexual comments:

> *It's hard to define what's a comment, and what's a sexual comment . . .*
>
> *I don't know if saying to someone, you look nice. . .*
>
> *It's nice to receive a compliment.*
>
> *I don't know if saying to someone, you look good, is a sexual comment.*

*[. . .]*

*I think it depends on the person, like someone might like getting nice comments about themselves, and then some people could say, it's rude, or something like that.*

(Sc2_S6B.2)

*Gender.* Behaviors were generally regarded as more acceptable if instigator and recipient were both the same gender. It was suggested that for making sexual jokes and name-calling, *"Sometimes boys are more likely to take it further than the girls [. . .] or try to embarrass"* (Sc1_S6G.2) so *"it makes it more awkward if a boy says it to you"* (Sc1_S2G). Reflecting on this, one girl-only discussion group suggested that boys and girls had different perspectives on what is funny and counts as just joking (Sc2_S4G) while another attributed it to differing levels of maturity (Sc1_S4G.1).

Some groups discussed gender double standards, presenting ideas which again appeared underpinned by beliefs around rights, this time the right to equal treatment. Thus, in respect of name-calling, girls suggested *"Boys get away with it [. . .] it's always, like, the girls that get the worst out of it"* (Sc1_S4G.1), while for showing or sending sexual pictures *"See if a girl and a boy does this and the picture gets sent around, the girl gets called a name, but the boy doesn't. He doesn't get called one thing"* (Sc2_S2G). However, boy-only discussion groups also gave examples of gender double-standards, working in the opposite way, particularly in respect of physical behaviors such as pulling at clothing:

*Everyone's done it. But that's like a boy to a pal, like another boy. To do it to a girl, even if they're your friend, it's still bad.*

*[. . .]*

(Sc2_S6B.2)

Girls were aware of this, with one older group suggesting that while a boy pulling at a girl's clothes, even if flirting, was not OK, it should therefore also be not OK for girls to target such behavior at boys, because it might make them feel uncomfortable (Sc2_S6G) and another that for all behaviors the gender of perpetrator and victim *"shouldn't matter"* (Sc1_S6G.1).

## Discussion

Our study was conducted against a background of very little research in respect of the prevalence of sexual harassment among UK adolescents and no identified recent studies internationally on adolescent perceptions of 'what counts' as sexual harassment. We have structured our discussion of findings around our key research questions.

### What is the self-reported prevalence of sexual harassment (victimization and perpetration) among young people?

Consistent with other studies, our survey found "sexual harassment is all too common for secondary school students" [63] (p343). Overall, around a third of our sample reported experiencing no sexual harassment at or on the way to school within the past three months, around a third visual/verbal harassment only and a third contact/personally-invasive harassment, with most of the latter group also reporting visual/verbal harassment. Unsurprisingly, rates of self-report perpetration were much lower, with over two-thirds reporting none, a fifth visual/verbal only and around one-in-twenty contact/personally-invasive perpetration. Although slightly

different methodologies mean direct comparisons are difficult, these results are broadly consistent with those of others [17, 22, 25, 35]. While the first (1993) Hostile Hallways survey found being shown or sent unwanted sexual images was the seventh most frequently experienced item [17], it was the second most frequently experienced in our survey, likely reflecting the increased ubiquity and ease of sharing such material online.

Every behavior was reported as having been experienced by some in our sample, although some were at very low levels. In addition, we found that for both victimization and perpetration, the behaviors we categorized as contact/personally-invasive, and which would generally be regarded as 'more serious', were unlikely to occur in the absence of 'lower level' visual/verbal behaviors. This suggests a hierarchy or gateway effect, such that contact/personally-invasive behaviors are experienced on top of more common visual/verbal behaviors, and that adolescents reporting contact/personally-invasive behaviors will have experienced (or perpetrated) a higher number of behaviors overall.

## What is the relationship between sexual harassment victimization and perpetration?

We found not just a strong relationship between sexual harassment victimization and perpetration, but also in respect of specific type, with almost all contact/personally-invasive perpetration occurring among those who had experienced contact/personally-invasive victimization. Associations between victimization and perpetration have been identified in respect of both bullying generally (e.g. [64]) and sexual harassment specifically; other studies have also found that many adolescent sexual harassment perpetrators are also victims [31], with a 2011 U.S study reporting that almost a quarter of perpetrators cited retaliation as a reason for their behavior [23]. There is also evidence that among adolescents, perpetration is more likely among those who have witnessed peers engage in more sexual harassment, and that homophily (within friendship group similarity) occurs in respect of potentially offensive sexual behaviors, suggesting the importance of perceived group norms and the potential for engaging in behaviors in order to fit in with the group [65, 66]

## How do sexual harassment victimization and perpetration differ according to gender, sexual orientation and age?

Sexual harassment is more frequently perpetrated by men against women, resulting in far higher rates among women over the course of a lifetime [67] and, consistent with this, studies generally find higher rates of victimization among young women and of perpetration among young men. We were therefore surprised to find very few gender differences in victimization. However, those we did find (shown/sent unwanted sexual material and pressured you to send them a naked/sexual picture more likely for girls and been flashed/mooned at more likely for boys) make intuitive sense. A previous study of adolescents which also did not find higher rates of victimization among girls actually found higher rates among boys, in particular that boys received more same-gender harassment than girls, with no difference in cross-gender harassment; we did not ask whether harassment was same or cross-gender. The authors of that study highlight the importance of not ignoring young men or assuming sexual harassment is solely a women's problem [26]. Our results in respect of gender differences in perpetration were as expected, with markedly higher rates among boys in respect of all 17 behaviors. The qualitative data were consistent in pointing towards greater perpetration and acceptability of, at least certain low-level verbal behaviors, by boys. Previous studies have generally found higher rates among young people whose sexual orientation is not heterosexual [15, 27–29]; our results were in line with this, although only for contact/personally-invasive (i.e. 'more serious')

victimization. While there may be no single cause of sexual harassment [67], seen through a feminist lens, it serves to perpetuate and reinforce hegemonic masculinities and dominant gender norms (heterosexuality). Adolescents may experience pressure to conform to these norms by either perpetrating or tolerating sexually harassing behaviors [63].

Findings regarding age differences in rates of both victimization and perpetration, particularly a clear increase in contact/personally-invasive types between ages 13 and 15 are also consistent with previous studies which have suggested a developmental explanation. Factors cited include pubertal changes, increases in sexual interest but not necessarily understandings of what is socially appropriate and a shift from same- to mixed-gender peer groups [26, 30].

## Are sexual harassment behaviors understood/labelled by young people as 'sexual harassment'?

Although most of the behaviors on our card task are included in standardized measures of sexual harassment such as Hostile Hallways, this was not the term which discussion of the cards generally brought to mind among these students. Nonetheless, they did label them pejoratively, and older students in particular understood the behaviors in terms of their non-consensual nature and impact on the recipient. Low-level verbal/visual behaviors were generally those that students were most likely to report being unsure about having experienced in the survey. Previous authors have suggested how such 'everyday sexism' is either unrecognized, normalized or perceived as not worth reporting [2, 63, 66, 68, 69]. Some view this as one end of a continuum of sexual violence in which (men's) 'typical' and 'aberrant' behaviors shade into one another [70, 71]. This highlights the importance of helping young people recognize how 'low-level' behaviors can play into norms and attitudes that support those that are more serious.

## What do young people consider acceptable and unacceptable behavior and what influences these perceptions?

The focus group discussions drew out nuance in how behaviors were perceived. The task prompted some fairly animated discussions, many concluding "it depends". There was almost no "it depends" in respect of behaviors which were perceived as coercive, non-consensual and persistent. Outside of this category, acceptability was influenced by two key dimensions: familiarity and relationship; and the instigator's (perceived) intent. The features identified by these young people were consistent with the few previous adolescent studies which have also recognized force, repetition, coercion, intent, personal targeting, familiarity and levels of discomfort as important criteria for judging acceptability [45, 49].

The results demonstrate the ways in which young people are required to make complex judgements as to how a certain set of factors might combine to make a behavior more or less acceptable: something which could be laughed off as obviously comedic if the instigator was a friend, might be perceived as hurtful if instigated by someone less trusted; different individuals might 'read' and therefore respond to the same behavior in different ways. It appeared that students were applying schemas, that is, knowledge, memories and behavioral exemplars to make sense of experiences [46]. These schemas were underpinned by two key factors. One was understandings about rights: legal rights; the rights of different groups to equal treatment; and the rights of others in relation to dignity and respect for their feelings and boundaries. Although reasoning and perspective-taking abilities advance during pre- and early adolescence, there can be very significant differences between young people of the same age, and development of these abilities generally continues into adulthood [72]. The second key factor was trust and 'knowingness' engendered by friendship. However the interacting nature of these schemas means that young people face tricky decisions in deciding a behavior's acceptability, with several

discussions identifying ambiguities and the possibility of making mistakes. As others have pointed out, this may be compounded, in adolescence, by pressures to develop sexual relations at the same time as learning to express desires appropriately [18, 40, 41].

## Strengths and limitations

Our study was restricted to only three schools (only two providing qualitative data) and had a lower survey response than might normally be expected in schools-based studies. However, it is unusual in collecting complementary quantitative and qualitative data and conducted in schools which varied in size, location, student socio-economic status and academic achievement. Our themes, identified inductively from the data, are consistent with those of others.

Like very many studies of the prevalence of sexual harassment in adolescence, we based our survey items on the original Hostile Hallways questionnaire which some have criticized as atheoretical and with limited psychometric information [14, 73]. Further, and also like many studies, we made some amendments, in particular slight modifications to encompass the possibility that certain behaviors might be online; this makes detailed comparisons with other studies difficult. In addition, our distinction between items representing visual/verbal and physical contact/personally-invasive harassment and perpetration was based on previous factor analyses of Hostile Hallways items [18, 19, 61] the results of which were not identical. Although others might therefore have chosen an alternative categorization, it is unlikely that this would have been markedly different and our categorizations were associated in expected ways with other variables, suggesting validity.

We chose to collect qualitative data via focus groups, not just because we believed that adolescents would be more comfortable discussing the acceptability of behaviors using this method rather than in individual interviews, but because we were interested in shared understandings and wanted to capture debate and the shaping of knowledge and norms via group interaction. It is important to acknowledge that our choice of group method means these are 'public' rather than 'private' data.

More qualitative data focused on jokes and name-calling, about which students were more likely to be able to speak with experience and in public than the less common (and so generally more hypothetical) and less acceptable behaviors. We are also aware that certain potentially important factors are missing from this analysis, including differences according to age, where issues were hinted at but data were insufficient to support fuller analysis.

## Conclusions and implications for interventions

Experience of a range of sexual harassment behaviors in the school setting, particularly being the recipient of sexual jokes, being shown unwanted sexual material and/or hurtfully described as gay or lesbian, is common and normalized in adolescence. A significant minority also experience 'more serious' behaviors such as sexual touching, blocking and/or being pressured to send naked or sexual pictures of themselves; harassment of this type is more likely among mid- or older adolescents and those whose sexual orientation is not heterosexual.

While adolescents perceive behaviors involving coercion as unacceptable, there are ambiguities around the acceptability of many behaviors generally regarded as representing sexual harassment which require complex judgements at a developmental stage marked by identity formation, exploration and initiation of intimate relationships. School-based interventions should recognize this by adopting an approach which not only aims to increase knowledge of sexual harassment, but also includes active learning, including discussions around, and challenges to, the factors underpinning young people's decisions on whether or not behaviors are acceptable, perhaps based on some of the ideas and statements included in our data. They

should also aim to develop understandings of how 'more serious', coercive and/or aberrant behaviors increase when 'less serious' behaviors are perceived as normal and acceptable, so feeding into 'rape culture'. In doing so, such interventions may expand the numbers, types and/or levels of behaviors understood as sexual harassment [46, 47] and disrupt its normalization [63], thus impacting both attitudes and perceived peer norms. Theories of reasoned action, which have been adopted to help explain many different types of behavior, including adolescent sexual harassment [74, 75], propose that behaviors are influenced by an individual's attitudes (e.g. is the behavior favorable, appropriate, etc) and perceived peer norms (would they engage and/or approve of others engaging in the behavior) [76]. This suggests that interventions which impact both attitudes and perceived peer norms via active learning, discussion and challenges would be more likely to impact on behaviors than those which focus only on increasing knowledge of sexual harassment.

The challenges for schools in addressing sexual harassment are significant. The 2021 UK government 'Review of sexual abuse in schools and colleges' recommends they aim for cultural change via a whole-school approach including via the curriculum, training of all staff, record-keeping, sanctions where appropriate, consistency in responses, close work with external safe-guarding partners and "better understand[ing of] the definitions of sexual harassment and sexual violence" [2]. In responding to this, school leaders and policy-makers should be cognizant of the complexity and ambiguities highlighted in our data. Adolescence presents a window of both opportunity and risk in respect of sexual harassment behaviors, during which an effective 'foot-in-the-door' intervention [77] could generate changes in attitudes and behaviors which persist into later life.

## Supporting information

**S1 Checklist. STROBE checklist of items that should be included in reports of *cross-sectional studies*.**
(DOC)

**S2 Checklist. Standards for reporting qualitative research checklist.**
(DOC)

**S1 File. Parent and student study information and consent forms.**
(DOC)

**S1 Table. Sexual harassment items—Self-reported victimization and perpetration—Descriptive data.**
(DOC)

**S2 Table. Composite variables: Basic frequencies.**
(DOC)

**S3 Table. Cross-tabulations of visual/verbal and contact/personally-invasive victimization and perpetration—Numbers (and valid cell percentages) and Chi-square.**
(DOC)

**S4 Table. Cross-tabulation of composite victimization and perpetration variables.**
(DOC)

## Acknowledgments

We would like to thank all the young people who participated in the Equally Safe at School study and the school staff who offered their time to facilitate this. We also acknowledge help

from MRC/CSO Social and Public Health Sciences Unit colleagues, including Drew Jackson (online questionnaire development and administration); Elaine Hindle (survey organization) and Abita Bhaskar (quantitative data management), as well as Kathryn Dawson and Laura Wylie, colleagues from Rape Crisis Scotland, who provided feedback on survey design.

## Author Contributions

**Conceptualization:** Helen Sweeting, Carolyn Blake, Kirstin R. Mitchell.

**Data curation:** Helen Sweeting, Carolyn Blake.

**Formal analysis:** Helen Sweeting, Carolyn Blake, Julie Riddell, Simon Barrett, Kirstin R. Mitchell.

**Funding acquisition:** Kirstin R. Mitchell.

**Investigation:** Helen Sweeting, Carolyn Blake, Julie Riddell, Simon Barrett, Kirstin R. Mitchell.

**Methodology:** Helen Sweeting, Carolyn Blake, Kirstin R. Mitchell.

**Project administration:** Helen Sweeting, Carolyn Blake, Kirstin R. Mitchell.

**Supervision:** Helen Sweeting, Carolyn Blake, Kirstin R. Mitchell.

**Writing – original draft:** Helen Sweeting.

**Writing – review & editing:** Helen Sweeting, Carolyn Blake, Julie Riddell, Simon Barrett, Kirstin R. Mitchell.

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
