## [Decision Letter · Decision Letter 0]

20 Dec 2021

Sexual harassment in secondary school: prevalence and ambiguities. A mixed methods study in Scottish schools

PONE-D-21-21275

Dear Dr. Helen Sweeting,

We’re pleased to inform you that your manuscript has been judged scientifically suitable for publication and will be formally accepted for publication once it meets all outstanding technical requirements.

Kind regards,

Shah Md Atiqul Haq

Academic Editor

PLOS ONE

Additional Editor Comments (optional):

Dear authors,

Your article is now accepted.

Reviewers' comments:

Reviewer's Responses to Questions

**Comments to the Author**

1. Is the manuscript technically sound, and do the data support the conclusions?

Reviewer #1: Yes

2. Has the statistical analysis been performed appropriately and rigorously? 

Reviewer #1: Yes

3. Have the authors made all data underlying the findings in their manuscript fully available?

Reviewer #1: Yes

4. Is the manuscript presented in an intelligible fashion and written in standard English?

Reviewer #1: Yes

5. Review Comments to the Author

Reviewer #1: This study explored prevalence and perceptions of sexual harassment in a school context. It utilised a mixed-methods approach study with 13-17 year-old adolescents which is highly commendable. The authors ensured diversity of their convenience sample by including schools of size, level of urbanisation, student socio-economic status and academic achievement. The findings are consistent with previous studies so that a large majority of adolescents are exposed to different forms of sexual harassment, but these are of lower intensity. The unwanted consequence is that they are normalized and often nor reported / reacted to. The more serious perpetrations, mainly physical, were found to be much less present. The fact that perpetrator are in many cases also victims, warrants further investigation.

This is a well designed, implemented and analyses study, and the manuscript is easy to read and follow.

6. PLOS authors have the option to publish the peer review history of their article (what does this mean?). If published, this will include your full peer review and any attached files.

Reviewer #1: **Yes: **Dean Ajdukovic

---

## [Editor Report · Acceptance letter]

2 Feb 2022

PONE-D-21-21275 

Sexual harassment in secondary school: prevalence and ambiguities. A mixed methods study in Scottish schools 

Dear Dr. Sweeting:

I'm pleased to inform you that your manuscript has been deemed suitable for publication in PLOS ONE. Congratulations! Your manuscript is now with our production department. 

Kind regards, 

on behalf of

Dr. Shah Md Atiqul Haq 

Section Editor

PLOS ONE